# Assessing the performance of a commercial multisensory sleep tracker

**Nanna J. Mouritzen** [1,2]*, **Lisbeth H. Larsen**[1], **Maja H. Lauritzen**[1], **Troels W. Kjær**[1,2]

**1** Department of Neurology, Center of Neurophysiology, Zealand University Hospital, Roskilde, Denmark,
**2** Department of Clinical Medicine, University of Copenhagen, Copenhagen, Denmark

* namou@regionsjaelland.dk

**Data Availability Statement:** According to Danish law, data cannot be shared publicly. However, data are available upon request from the Research Unit, Center of Neurophysiology, Zealand University Hospital (email: suh-neu-kfe@regionsjaelland.dk)

## Abstract

Wearable sleep technology allows for a less intruding sleep assessment than PSG, especially in long-term sleep monitoring. Though such devices are less accurate than PSG, sleep trackers may still provide valuable information. This study aimed to validate a commercial sleep tracker, Garmin Vivosmart 4 (GV4), against polysomnography (PSG) and to evaluate intra-device reliability (GV4 vs. GV4). Eighteen able-bodied adults (13 females, *M* = 56.1 ± 12.0 years) with no self-reported sleep disorders were simultaneously sleep monitored by GV4 and PSG for one night while intra-device reliability was monitored in one participant for 23 consecutive nights. Intra-device agreement was considered sufficient (observed agreement = 0.85 ± 0.13, Cohen's kappa = 0.68 ± 0.24). GV4 detected sleep with high accuracy (0.90) and sensitivity (0.98) but low specificity (0.28). Cohen's kappa was calculated for sleep/wake detection (0.33) and sleep stage detection (0.20). GV4 significantly underestimated time awake (*p* = 0.001) including wake after sleep onset (WASO) (*p* = 0.001), and overestimated light sleep (*p* = 0.045) and total sleep time (TST) (*p* = 0.001) (paired *t*-test). Sleep onset and sleep end differed insignificantly from PSG values. Our results suggest that GV4 is not able to reliably describe sleep architecture but may allow for detection of changes in sleep onset, sleep end, and TST (ICC ≥ 0.825) in longitudinally followed groups. Still, generalizations are difficult due to our sample limitations.

## 1. Introduction

Insufficient sleep constitutes a large health problem. In 2017, 46% of the Danish population reported sleep problems, and the prevalence has been increasing [1]. Sleep diaries are a common way to assess sleep. Though sleep diaries are convenient, non-intrusive, and allow for subjective sleep assessment, they do not describe sleep architecture, and erroneous sleep estimates may be reported [2]. In long-term monitoring, sleep diaries may be further encumbered by compliance problems, providing missing data.

Objective sleep assessment is ideally obtained with polysomnography (PSG), considered the gold standard. During sleep, specific physiological signals are recorded allowing for the differentiation between sleep stages and calculation of sleep parameters. A central problem when using PSG is the time-consuming and resource-demanding factor requiring expensive equipment and specialized personnel. Thus, PSG is not suitable for long-term sleep monitoring. The

for researchers who meet the criteria for access to confidential data.

**Funding:** The project was funded by the EU Interreg Deutschland-Danmark (087-1.1-18) and a pre-graduate research scholarship was awarded by the Lundbeck Foundation to NJM.

**Competing interests:** Garmin® kindly borrowed us GV4 watches to perform our experiment. This had no influence on our study design, data analysis, results or conclusions and does not alter our adherence to the PLOS ONE policies on sharing data and materials.

need for a practical sleep monitoring device is reflected in the rapid development of commercial activity trackers that enables self-monitoring of various biomarkers, including sleep parameters [3]. Almost 30% of US respondents reported to self-monitor one or more health parameters with an application, a band, clip, or smartwatch in 2016 [4]. Wrist-borne consumer sleep trackers are primarily accelerometer-based and assess sleep similar to actigraphy where movement denotes wake and lack of movement denotes sleep.

The great adoption and development of sleep trackers are accompanied by a growing demand for validation to make use of the provided sleep information. The majority of sleep trackers detect sleep with high sensitivity but low specificity [5, 6]. Mostly, they overestimate total sleep time (TST) and sleep efficiency (SE), and underestimate wake including sleep onset latency (SOL) and wake after sleep onset (WASO) [7–9]. These estimation biases are also known from standard actigraphs [10, 11]. Similar to actigraphy, more pronounced discrepancies have been implied in older and sleep-disrupted populations [9, 12, 13]. Estimation biases of sleep stage durations are not marked by a clear tendency [14–16], but current sleep trackers describe sleep architecture poorly. Though often overestimated, TST is repeatedly emphasized as the most reliable parameter calculated by sleep trackers [9, 14, 15, 17].

Autonomic changes accompany transitions between sleep stages. Measures of such physiological changes have been employed to aid accelerometer-based sleep scoring (the multisensory approach). When falling to sleep, parasympathetic influence gradually increases through non rapid eye movement sleep (NREM) while sympathetic influence decreases. This results in reduced cardiac activity. When progressing into rapid eye movement sleep (REM), and specifically into phasic REM, sympathetic influence increases together with cardiac activity [18, 19]. Optical photoplethysmography (PPG) is a mechanism that detects changes in blood volume and thereby measures heart rate (HR) and HR variability (HRV) by illuminating the skin with subsequent registration of changes in the reflected light. HRV frequencies can help indicate shifts between sleep stages. The newer generation of sleep trackers apply both accelerometer and PPG data in sleep scoring, but the beneficial role of additional PPG information is ambiguous.

New sleep tracking technology must result in improved performance and for that reason, the validity of potentially beneficial devices should be explored. This study aims to evaluate the validity of sleep/wake and sleep stage detection provided by a wrist-borne, accelerometer, and PPG based device, Garmin Vivosmart 4 (GV4). As PSG is considered the gold standard, GV4 is compared against PSG (inter-device reliability). The applied PPG feature is expected to improve GV4's ability to distinguish between sleep stages and to detect wake periods. GV4 vs. GV4 (intra-device) reliability is also investigated because sufficient reliability between identical GV4 devices is considered a prerequisite for validation of GV4 against gold standard.

## 2. Methods

### 2.1 Participants and study design

This is a validation study with measurements of sleep parameters from one night of simultaneous PSG and GV4 sleep monitoring for each participant. This study is a sub-study of an ongoing longitudinal cross-over study from which participants ($N = 18$) were invited to additional sleep testing. Participants were required to be $\geq 18$ years old, able-bodied and without any self-reported previous history of clinically diagnosed sleep disorders. The trial protocol was approved by the Ethics Committee of Region Zealand (protocol number: SJ-780).

### 2.2 Materials

**2.2.1 Polysomnography.** Overnight home PSG (AASM Type II study) was obtained by use of TrackIT™ Mk3 Sleep Click-on 2 (Lifelines, Neuro). Sleep was unattended without

monitoring by a technician or by video. Participants were equipped with eight scalp electrodes for EEG recording (providing F3/M2, F4/M1, C3/M2, C4/M1, O1/M2 and O2/M1 derivations), two electrodes at the chin recording EMG, two electrodes near the eyes recording EOG, and two electrodes on the chest recording a lead II ECG. All electrodes were positioned in accordance with the American Academy of Sleep Medicine (AASM) guidelines. Recordings of airflow and respiratory effort were left out since the study focused only on sleep staging and its associated measures. Participants were asked to press an event marker button on the PSG device when going to bed (identified as "lights off"). This mark represents the time when the participant intends to sleep but does not sleep yet. Recordings were initiated before the participants went home and terminated when they came in the next day to deliver the detached PSG equipment. Afterwards, the AASM Scoring Manual Version 2.2 was used to evaluate PSG files in epochs of 30 s in the Nicolet One 5.95 software by an experienced technician (MHL). Lastly, all PSG recordings were reviewed by a board-certified neurophysiologist (TWK).

**2.2.2 Garmin Vivosmart 4.** GV4 is a commercially available, off-the-shelf activity watch. Besides possessing the ability to measure the daily number of steps, HR, oxygen saturation, etc., it claims to have a sleep tracking function. This function is based on limb movements, detected via an embedded triaxial accelerometer, and PPG signals. On this basis, an algorithm attempts to detect sleep stages and wake depending on threshold values [5, 20]. The collected sleep information include sleep onset, end, duration, stages (light, deep, REM, and wake), and level of movement during sleep [21]. Due to the pre-formed device settings, data were collected in epoch lengths of 1 min. PSG and GV4 data were collected from December 2019 until March 2020.

## 2.3 Procedure

Intra-device reliability was examined by equipping one participant with two identical GV4 devices for 23 nights. In order to perform optimally, GV4 needs to adjust to the user, lasting ~14 days (personal communication). The devices were connected to two separate and newly created Garmin Connect™ accounts, eliminating potential influence from earlier logged data. Each software was updated, and the watches were set up identically and synchronized with their accounts. Both devices were worn on the non-dominant wrist side-by-side, and device position (proximal, distal) was shifted each night.

As part of the larger study, written informed consent was obtained from all participants. Participants were measured (height, weight) and had filled out a questionnaire prior to sleep testing including questions about their sleep. Participants were prepared for simultaneous GV4 and PSG sleep monitoring as earlier described. Next morning, sleep data from GV4 were extracted via the participant's online Garmin Connect™ account.

## 2.4 Data analysis

Our analyses focused on intra-device and inter-device reliability. For both of them, descriptive statistics were calculated using Microsoft Excel (2019) and MATLAB (R2019a).

GV4 and PSG data were checked for normality, and paired $t$-tests were performed to identify significant differences in sleep estimates. We used 95% confidence intervals and $p$-values < 0.05 were considered statistically significant. To evaluate sleep stage agreement, epoch-by-epoch (EBE) analysis was performed in both intra-device and inter-device comparisons. Observed agreement ($P_O$) and Cohen's kappa ($\kappa$) were calculated based on the confusion matrices (Tables 2 and 4).

**2.4.1 Intra-device reliability.** EBE analysis was performed by aligning and comparing data files from both GV4 devices in 1 min epochs (default). Sleep scorings were compared

Table 1. Demographic characteristics of participants.

| Variable (N = 18) | Mean ± sd |
|---|---|
| Age, years | 56.1 ± 12.0 |
| Sex, female | 13 (72.2%) |
| Body mass index, kg/m$^2$ | 27.6 ± 5.0 |
| Sleep problems within last 14 days | 3 (16.7%) |

from the time when the first watch detected sleep until the time when both watches detected one wake epoch the next morning. Comparisons of sleep measures were based on these time intervals as were comparisons of HR (bpm) and accelerometer data ("level of movement") to identify the source of a potential disagreement.

**2.4.2 Inter-device reliability.** To compare the GV4 data files with the corresponding PSG data files, GV4 epochs were doubled to match the 30 s PSG epoch resolution. Thus, a 1 min GV4 epoch, e.g. staged as light sleep, was extended to two epochs of 30 s, both staged as light sleep. This was done in accordance with former comparison studies [9, 22]. Both GV4 and PSG were manually synchronized using the internet clock time before each PSG study. GV4 and PSG data were compared from "lights off", registered by the participant, until the first wake epoch registered by both GV4 and PSG (scored by MHL). If GV4 sleep onset fell before the PSG event mark, we initiated the comparison from GV4 sleep onset.

Because some participants reported earlier sleep onset detected by GV4 compared to their own experience (confirmed by preliminary PSG results), we attempted to by-pass a potential misalignment by comparing data differently. This was approached using three different methods: The first comparison was based on conventional time alignment and the second on "sleep alignment", comparing data from the first registered sleep epoch in both methods (independent of time). The third was based on cross correlation, displacing GV4 data 40 epochs forward and backward, relative to PSG, from the time aligned point ($P_O$ was evaluated for each 30 s displacement).

Accuracy (proportion of true sleep and true wake epochs), sensitivity (proportion of true sleep epochs), specificity (proportion of true wake epochs), $P_O$ (proportion of correctly sleep staged epochs), and Cohen's kappa were calculated as mean values from the EBE analyses performed in each participant. Those values were also calculated for each type of alignment. We have reported the highest values of Cohen's kappa, achieved from one overall matrix containing all epochs (Table 4). Confusion matrices were used to identify misclassifications. To quantify the influence of the artificial extension of the original GV4 1 min epochs into 2 x 30 s epochs, performance values were additionally calculated only when two equal PSG epochs followed each other (i.e. evaluation of GV4 epochs were only made when PSG showed a similar data resolution equivalent to 2 x 30 s).

Table 2. Confusion matrix (intra-device agreement).

| GV4 1 | GV4 2 | | | | |
|---|---|---|---|---|---|
| Epoch | Wake | Light sleep | Deep sleep | REM sleep | Subtotal |
| Wake | 98 | 95 | 60 | 12 | 265 |
| Light sleep | 119 | 7266 | 504 | 107 | 7996 |
| Deep sleep | 41 | 701 | 2650 | 13 | 3405 |
| REM sleep | 4 | 141 | 16 | 166 | 327 |
| Subtotal | 262 | 8203 | 3230 | 298 | 11993 |

Values are based on 1 min epochs from 23 nights and presented as frequencies.

**Table 3. Single comparisons of sleep parameters between PSG and GV4.**

| Sleep parameter | PSG | GV4 | Mean difference (CI) | $p$ | $R^2$ | ICC |
|---|---|---|---|---|---|---|
| TST, min | 414.9 ± 69.4 | 442.8 ± 59.7 | 27.8 ± 29.5 (14.2 to 41.5) | 0.001* | 0.822 | 0.825 |
| TIB, min | 455.0 ± 65.3 | 453.2 ± 63.5 | -1.8 ± 26.2 (-14.0 to 10.3) | 0.771 | 0.842 | 0.921 |
| Wake, min | 50.3 ± 28.7 | 23.0 ± 18.6 | -27.3 ± 28.9 (-40.7 to 13.9) | 0.001* | 0.098 | 0.179 |
| WASO, min | 29.6 ± 24.0 | 6.6 ± 6.7 | -23.0 ± 23.2 (-33.7 to 12.3) | 0.001* | 0.067 | 0.074 |
| Light sleep (N1+N2), min | 223.3 ± 42.9 | 259.8 ± 79.5 | 36.5 ± 71.7 (3.4 to 69.6) | 0.045* | 0.196 | 0.328 |
| Deep sleep (N3), min | 93.7 ± 47.1 | 107.1 ± 91.9 | 13.4 ± 98.1 (-31.9 to 58.8) | 0.570 | 0.014 | 0.100 |
| REM sleep, min | 97.9 ± 33.0 | 75.8 ± 49.3 | -22.1 ± 54.7 (-47.4 to 3.2) | 0.105 | 0.026 | 0.136 |
| SOL, min | 10.4 ± 10.8 | 3.8 ± 24.3 | -6.7 ± 24.6 (-18.0 to 4.7) | 0.265 | 0.039 | 0.144 |
| REM latency, min | 73.5 ± 14.8 | 89.6 ± 34.7 | 16.1 ± 40.3 (-5.1 to 37.2) | 0.160 | 0.039 | -0.129 |
| SE, % | 90.6 ± 6.8 | 97.9 ± 5.2 | 7.3 ± 7.9 (3.6 to 10.9) | 0.001* | 0.020 | 0.081 |
| Sleep onset, min | 1405.6 ± 73.7 | 1399.1 ± 73.7 | -6.5 ± 24.6 (-17.8 to 4.9) | 0.280 | 0.892 | 0.944 |
| Sleep end, min | 409.0 ± 66.8 | 409.4 ± 66.9 | 0.4 ± 18.3 (-8.0 to 8.8) | 0.927 | 0.927 | 0.965 |

Data are expressed as means ± sd values. Total sleep time (TST), time in bed (TIB), wake after sleep onset (WASO), sleep onset latency (SOL).

*$p < 0.05$.

Sleep estimates measured by single modalities were compared (Table 3). GV4 "light sleep" was compared to PSG N1 + N2 sleep, and GV4 "deep sleep" to PSG N3 sleep. Presented GV4 measures of SOL, REM latency, TIB, and SE were artificially constructed subsequent to data extraction (GV4 does not automatically generate these measures). SOL was calculated as the time from the PSG event mark until GV4 sleep onset, REM latency as the time from GV4 sleep onset until the first GV4 detected epoch of REM, TIB as the sum of SOL, WASO and TST, and SE as TST divided by TIB. These measures were calculated for clinical interest. "Wake" included wake periods before sleep onset (from the first compared epoch) and after sleep onset until both methods agreed on sleep end with 1 wake epoch.

Linear regression was performed on each relationship between GV4 and PSG estimates providing $R^2$ values. Intraclass correlations (ICC) for sleep parameters were calculated using IBM SPSS Statistics (version 27) and were based on a single-rater, absolute-agreement, two-way random model.

**Table 4. Confusion matrices (inter-device agreement).**

| | | | | (A) | | | |
|---|---|---|---|---|---|---|---|
| PSG | | | | GV4 | | | |
| | Epoch | Wake | Light sleep | Deep sleep | REM sleep | Subtotal | |
| | Wake | 501 | 851 | 187 | 270 | 1809 | |
| | Light sleep | 297 | 4878 | 1696 | 1166 | 8037 | |
| | Deep sleep | 8 | 1709 | 1526 | 130 | 3373 | |
| | REM sleep | 21 | 1896 | 430 | 1150 | 3497 | |
| | Subtotal | 827 | 9334 | 3839 | 2716 | 16716 | |
| | | | | (B) | | | |
| PSG | | | | GV4 | | | |
| | Epoch | Wake | | Sleep | | Subtotal | |
| | Wake | 501 | | 1308 | | 1809 | |
| | Sleep | 326 | | 14581 | | 14907 | |
| | Subtotal | 827 | | 15889 | | 16716 | |

Values are presented as frequencies. A) is based on four sleep stages and B) is based on sleep/wake.

Bland-Altman plots were used to visualize the agreement between PSG and GV4 of sleep parameters and to identify potential patterns in biases. Limits of agreement (± 1.96 sd) were additionally calculated and applied.

## 3. Results

### 3.1 Missing data

We had an extremely high compliance of 99.51%–calculated as all unlabeled epochs divided by the total number of epochs. GV4 accounted for 0.19% and PSG for 0.30% of the 0.49% of missing data. The intra-device reliability data set had no missing data.

### 3.2 Demographics

Demographic characteristics of the study population are presented in Table 1. 61.1% of participants were moderate to severely overweight (BMI $\geq$ 25 kg/m$^2$) and 16.7% reported to have experienced mild sleep problems within the preceding 14 days. None reported any severe sleep problems.

### 3.3 Intra-device reliability

We calculated the observed agreement, $P_O$ = 0.85 ± 0.13 ($0.80 \leq P_O \leq 0.90$), and Cohen's kappa, $\kappa$ = 0.68 ± 0.24 ($0.58 \leq \kappa \leq 0.77$), from a confusion matrix (Table 2) based on epochs from 23 nights and two GV4 watches.

Two successive nights showed exceptionally low levels of agreement ($\kappa$ = -0.01 and 0.02). Agreement was $\geq$ 85% for HR and $\geq$ 96% for movement during these nights (proportion of agreement ± 1 unit). When reliability measures of the two nights were excluded, observed agreement and Cohen's kappa reached 0.89 ± 0.05 and 0.75 ± 0.11, respectively. No significant intra-device differences were noticed in the durations of TST, wake, light, deep or REM sleep ($p$ > 0.23 of all mentioned sleep outcomes). Device position did not affect agreement between devices ($p$ > 0.85 for either $P_O$ or $\kappa$). We assumed an identical calibration process during the first 14 days, due to identical device exposures, and therefore, we included all 23 nights in our calculations.

### 3.4 Inter-device reliability

**3.4.1 Paired comparisons.** Sleep parameters measured by PSG and GV4 and differences between the single modalities are shown in Table 3. For four participants, REM latency was not possible to calculate because GV4 ($N$ = 3) or PSG ($N$ = 1) did not detect any REM sleep. Therefore, REM latency is based on $N$ = 14. Also, GV4 initiated sleep detection before the PSG event mark button was pressed in four participants, resulting in a negative SOL and a SE > 100%, consequently, lowering and increasing mean values, respectively. Sleep onset and sleep end are expressed in minutes from 00:00:00 (HH:MM:SS).

Significant differences included overestimations of TST by 27.8 min, SE by 7.3% and light sleep by 36.5 min, and underestimations of wake by 27.3 min and WASO by 23.0 min.

**3.4.2 Bland-Altman analysis.** Fig 1 shows Bland-Altman plots for TST and WASO with lower and upper limits of agreement (± 1.96 sd). Positive mean differences denote GV4 underestimation of sleep parameters and vice versa as in previous validation studies [12, 14, 23, 24]. The plots confirm the obtained results from the paired *t*-tests, visualizing biases. General overestimations of TST and increasing discrepancy between methods in participants with higher amounts of WASO are illustrated. For both, measurements from three participants fell outside the limits of agreement out of which two participants were the same.

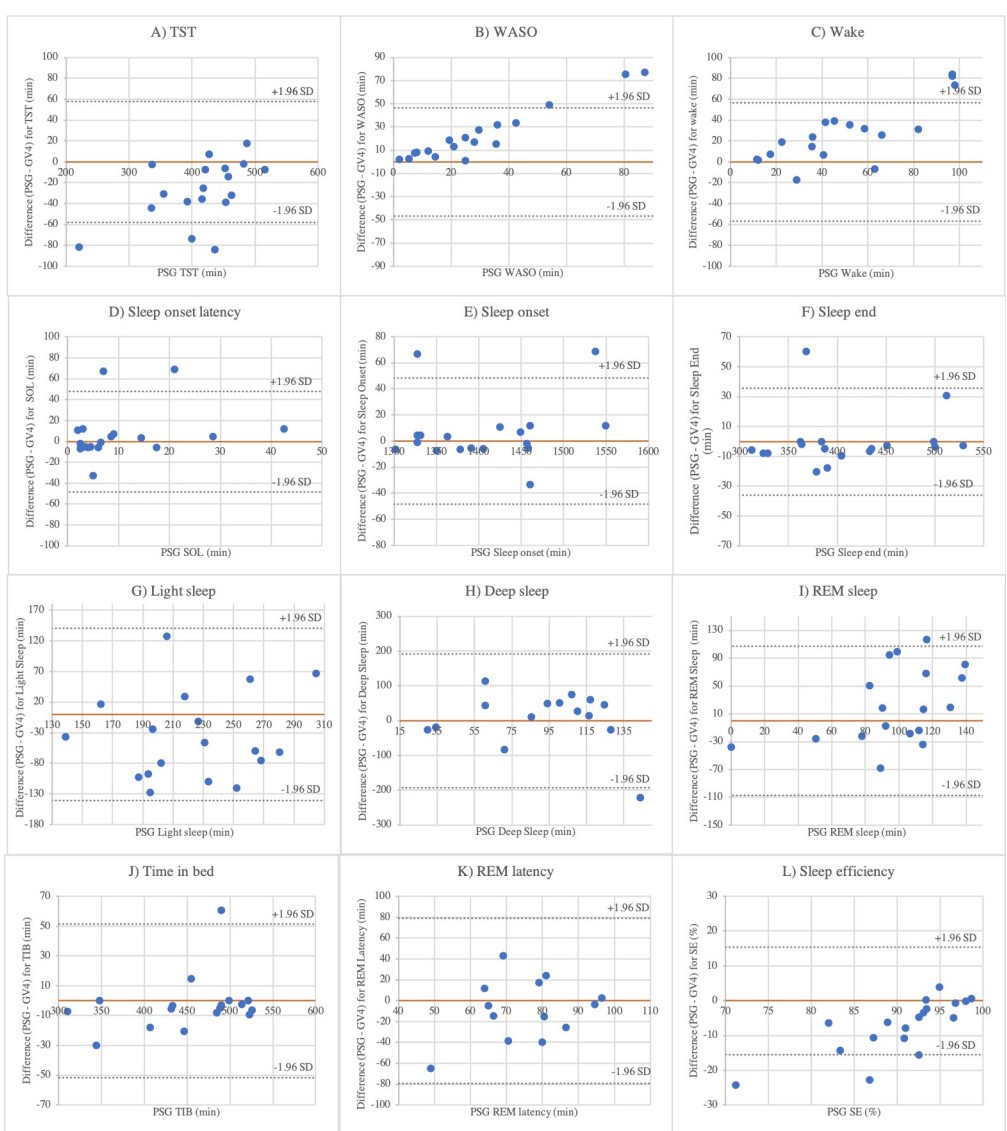

**Fig 1.** Bland-Altman plots for agreement between PSG and GV4 measured TST (A) and WASO (B).

**3.4.3 Linear regression analysis.** Fig 2 illustrates the relationships between PSG and GV4 measured sleep onset and sleep end. The coefficients of determination, $R^2$, indicate linearity in both cases.

**3.4.4 Epoch-by-epoch analysis.** Based on the confusion matrices, sensitivity was calculated to 0.98 ± 0.03, specificity to 0.30 ± 0.17, accuracy to 0.90 ± 0.06, observed agreement to 0.48 ± 0.10, and Cohen's kappa to 0.20 ± 0.11 for all sleep stages and 0.33 ± 0.18 for sleep/wake. Sensitivity for light, deep, and REM sleep were calculated to 0.60 ± 0.19, 0.45 ± 0.26, and 0.34 ± 0.26, respectively. Specificity for light, deep and REM sleep were calculated to 0.49 ± 0.13, 0.83 ± 0.20, and 0.88 ± 0.08, respectively.

No remarkable differences in performance values arose between the three alignment approaches and for that reason, the conventional time alignment was chosen to calculate the final results from the EBE analysis.

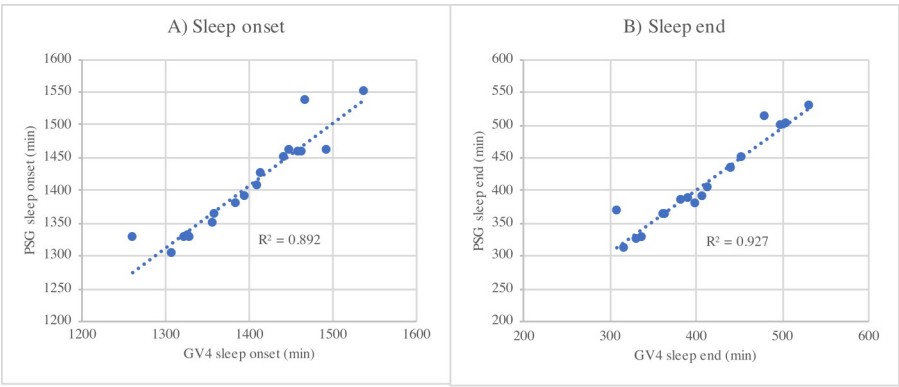

**Fig 2.** Simple linear regression analyses of sleep onset (A) and sleep end (B). $R^2$ values are shown.

Since we extended GV4's default 1 min epochs into 2 x 30 s epochs, performance could be falsely diluted. When we examined this potentially negative effect, calculating performance values based on comparison only when two equal PSG epochs followed each other, we found no noteworthy difference.

## 4. Discussion

### 4.1 Main findings

Evaluation of GV4 validity demonstrated sufficient intra-device agreement, and specific inter-device measures were reliably estimated. Sleep onset and sleep end were highly correlated with and differed insignificantly from PSG measures. TST was significantly overestimated but also highly correlated to TST measured by PSG. Though sleep architecture was poorly described by GV4, changes in sleep onset, sleep end and TST may be reliably detected in group settings over time.

### 4.2 Intra-device reliability

Values for intra-device reliability were lower than expected but comparable to the overall sleep stage agreement found between human scorers on different experience levels (= 0.83) [25] and to the inter-scorer reliability found between Chinese and US doctors (κ = 0.75 ± 0.01) [26]. According to Landis and Koch's arbitrary divisions [27], our kappa statistic indicates a substantial agreement across GV4 devices. The relatively low intra-device agreement may be due to differences in software (data processing) rather than hardware (sensors) since high agreement was found between devices in the detection of both HR and movement.

One study found high agreement between Fitbit devices (96.5–99.1%) [22], evaluating only three nights while we evaluated 23. Another study found no significant differences between Fitbit devices in SE and TST estimations agreeing with our findings [17].

### 4.3 Estimation accuracy

Poor sleep increases the risk of concentrating problems, traffic accidents, overweight, lifestyle diseases, impaired immune function, and even mortality [28]. Therefore, high estimation accuracy is important to detect poor sleep and changes in sleep over time.

GV4 significantly underestimated wake, including WASO, and significantly overestimated TST, SE, and light sleep (Table 3). This is in accordance with previous results in other commercial sleep trackers [5]. Although TST was significantly overestimated, primarily due to poor wake detection, it is strongly correlated to TST measured by PSG. Of note, a study

comparing various sleep trackers with PSG also found this strong correlation between TST measurements ($r \geq 0.84$) [14].

In the present study, most frequent misclassifications were deep as light sleep (50.7%) and vice versa (21.1%), wake as light sleep (44.1%), and REM as light sleep (54.2%) matching earlier findings [16, 29] (Table 4A). This was in accordance with GV4 light sleep overestimations. Interestingly, misclassifications between light and deep sleep, and wake and light sleep correspond to typical misclassifications found between experienced human sleep scorers [26].

We used GV4 rather than a standard accelerometer since PPG signals are included in the sleep scoring algorithm [30], supposed to improve sleep scoring. The finding of a low kappa value indicates poor sleep stage differentiation in spite of included PPG signals. However, the beneficial role of PPG signals in accelerometer-based sleep tracking remains unilluminated, since we do not know how these signals are processed and applied.

The uneven distribution of epochs in the marginal totals with a symmetrical high amount of sleep epochs and low amount of wake epochs (Table 4), causes a higher chance-expected agreement. This translates into a lower kappa value though observed agreement is high, described by Feinstein and Cicchetti as "the first paradox of kappa" [31].

Peculiarly, a preliminary validation study of Garmin Vivosmart 3 ($N = 55$) has presented a Cohen's kappa as high as $0.54 \pm 0.12$ [20] using a three-channel EEG system as reference method. Unfortunately, sparse information is available, obstructing comparison with our study. Nevertheless, a study performed in "good sleepers" ($N = 17$) [9], comparing Fitbit against PSG, calculated a Cohen's kappa of 0.38, close to our kappa value.

## 4.4 Sleep/wake detection

The intrinsic sleep/wake detection sensitivity of GV4 causes a high sensitivity at the cost of a low specificity. This agrees with previous studies [9, 16, 32]. Classification of sleep/wake in GV4 is based on movement and HRV. Lack of movement far from always specifies sleep, and motionless wake has been acknowledged as a huge factor hampering wake detection [5].

Though GV4 SOL was artificially constructed, the range of the PSG device difference included negative values indicating a risk of underestimation (Table 3). In four participants, the event marker button was pushed after GV4 initiated sleep detection, underlining its insufficient wake identification. Furthermore, PSG device discrepancy increased almost constantly in participants with a higher content of WASO (Fig 1B). WASO measured by PSG had a notably larger sd than WASO measured by GV4 (Table 3). This underlines a general tendency with underestimations of WASO across participants. Therefore, GV4 may be less erroneous to measure WASO in individuals with lower amounts.

The high average age of our participants as well as the "first-night effect" may provide a more fragmented sleep pattern in the sample, lowering specificity [5, 12, 33]. We performed home PSG which is known to reduce the first night effect [34]. Only 16.7% of our participants had PSG SE < 85% (SE $\geq$ 85% is considered "normal" according to the Pittsburg Sleep Quality Index [35]). Thus, we do not believe poor sleep caused poorer device performance.

Registration of habitual bedtimes and awakening times are required by GV4. Unprecise adjustment can supposedly affect performance as can a too short calibration period. However, poorer performance was not observed in participants with inaccurate adjustments and all of the participants had worn the GV4 > 14 days, ensuring sufficient calibration.

## 4.5 Limitations

Generalization of our results is limited by the small sample size with an unequal distribution of gender and age.

Because no consensus exists about to which extent new sleep technology should perform, sleep tracker comparisons are challenging due to the unstandardized use of methods and performance measures. Formulation of device standards would aid the evaluation and comparison of future sleep trackers.

The proprietary nature of commercial sleep tracker algorithms encumbers their validation. Furthermore, is the use of commercial sleep trackers accompanied by a risk of software updates that without warning can change the sleep scoring algorithms.

### 4.6 Future perspectives

Interpretation of the provided sleep information by GV4 should be done with caution most importantly because GV4 poorly describes sleep architecture. Thus, use of GV4 should be avoided when exact estimations of sleep parameters are essential. However, when PSG is unsuitable, GV4 may be an interesting sleep assessment tool. In both the intra-device and inter-device comparisons, agreement was high for most nights with only a few outliers. Performance is thus considered acceptable when use is intended in groups over longer periods, rather than in individuals over shorter periods.

In summary, our results confirmed that GV4 is not able to reliably describe sleep architecture but may accurately detect changes in sleep onset, sleep end, and TST though generalizations are difficult due to our sample limitations. Thus, our findings should be confirmed by further studies. However, this information contributes to the field of sleep monitoring, especially for research purposes. GV4's abilities can impart objective sleep information, as we know sleep reporting methods like sleep diaries can induce problems of compliance in longitudinal study designs.

## Acknowledgments

The authors thank Sirin W. Gangstad for providing MATLAB scripts, facilitating our data analyses.

## Author Contributions

**Conceptualization:** Lisbeth H. Larsen, Troels W. Kjær.

**Data curation:** Nanna J. Mouritzen, Lisbeth H. Larsen, Maja H. Lauritzen.

**Formal analysis:** Nanna J. Mouritzen, Lisbeth H. Larsen, Maja H. Lauritzen, Troels W. Kjær.

**Funding acquisition:** Lisbeth H. Larsen, Maja H. Lauritzen, Troels W. Kjær.

**Investigation:** Nanna J. Mouritzen, Lisbeth H. Larsen, Maja H. Lauritzen, Troels W. Kjær.

**Methodology:** Nanna J. Mouritzen, Lisbeth H. Larsen, Maja H. Lauritzen, Troels W. Kjær.

**Project administration:** Lisbeth H. Larsen, Maja H. Lauritzen, Troels W. Kjær.

**Resources:** Lisbeth H. Larsen, Maja H. Lauritzen, Troels W. Kjær.

**Supervision:** Lisbeth H. Larsen, Maja H. Lauritzen, Troels W. Kjær.

**Validation:** Nanna J. Mouritzen, Lisbeth H. Larsen, Maja H. Lauritzen, Troels W. Kjær.

**Visualization:** Nanna J. Mouritzen.

**Writing – original draft:** Nanna J. Mouritzen.

**Writing – review & editing:** Nanna J. Mouritzen, Lisbeth H. Larsen, Maja H. Lauritzen, Troels W. Kjær.

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
