## [Decision Letter · Decision Letter 0]

4 Aug 2020

PONE-D-20-21216

Validation of a Commercial Multisensory Sleep Tracker

PLOS ONE

Dear Dr. Mouritzen,

Thank you for submitting your manuscript to PLOS ONE. After careful consideration, we feel that it has merit but does not fully meet PLOS ONE’s publication criteria as it currently stands. Therefore, we invite you to submit a revised version of the manuscript that addresses the points raised during the review process.

Please note that both reviewers exoressed a series of concerns that must all be addressed.

We look forward to receiving your revised manuscript.

Kind regards,

Raffaele Ferri, MD

Academic Editor

PLOS ONE

Journal Requirements:

2. Thank you for including your ethics statement: 'The trial protocol was approved by the Regional Health Research Ethics Committee (SJ-780).'

3. Thank you for including your competing interests statement; "The study was based on Garmin Vivosmart 4 sleep tracking because of its multisensory technology and user-friendly design. Garmin kindly borrowed us Garmin watches to perform the experiment. This had no influence on the study design, data collection or analysis, decision to publish, or preparation of the manuscript."

Reviewers' comments:

Reviewer's Responses to Questions

**Comments to the Author**

1. Is the manuscript technically sound, and do the data support the conclusions?

Reviewer #1: Yes

Reviewer #2: Partly

2. Has the statistical analysis been performed appropriately and rigorously? 

Reviewer #1: Yes

Reviewer #2: N/A

3. Have the authors made all data underlying the findings in their manuscript fully available?

Reviewer #1: Yes

Reviewer #2: No

4. Is the manuscript presented in an intelligible fashion and written in standard English?

Reviewer #1: Yes

Reviewer #2: Yes

5. Review Comments to the Author

Reviewer #1: This is a validation study for a commercial sleep tracker from Garmin. The study was carried out in parallel with polysomnography, which is the reference standard for sleep recording. A small number of subjects was investigated: 18 healthy persons. One person was recorded for 23 nights, but these were not with polysomnography. The results show that the sleep tracker is good for sleep onset and end of sleep, but not good for sleep stages. This appears to be similar to actigraphy which has not been tested here.

As written in the ‘Polysomnography’ section, I assume that this is not a AASM type I study with supervised polysomnography, but a AASM type II study with polysomnography at home. If this is the case, please state this clearly, because this setting is different to most settings. In type II studies, there is no video recording or no monitoring from an experienced sleep technician. This needs to be noted.

For the smartwatch, you say that it considers not only actigraphy, but PPG as well. But how is this done? Since the results show, that these are comparable to simple actigraphy, may be PPG is not really evaluated? If we consider the WatchPAT device from Itamar as a smartwatch as well, then that device does a better job in evaluationg the PPG signal. Please compare critically.

I am especially interest how the Garmin evaluates REM sleep. That it evaluates REM sleep is only mentioned in brackets in the methods and details are only given when one reads the results. I think this capability of Garmin to detect REM sleep needs to be described in the methods section.

I think it is very good to check for 23 consecutive nights and it is very good to have two devices simultaneously.

As you performed an epoch by epoch comparison, it is of interest, how you arranged for a good synchronization between the polysomnography and the Garmin device. How much was the synchronization error then, less than a second?

The critical summary is appreciated.

Reviewer #2: Authors aimed at evaluating the performance of GV4 against PSG in measuring several sleep parameters in a small sample of 18 “healthy” adults in their home environment. I agree with the authors that the validation of such devices is an important step for adoption. The manuscript is well written.

Comments:

- Please consider the guidelines outlined by Depner at al., 2020 - Wearable Technologies for Developing Sleep and Circadian Biomarkers A Summary of Workshop Discussions

- I would use the term “assessing the performance” instead of “validating”.

- “without sleep disorder”. How sleep disorders were evaluated? Self-report?

- Not sure about having the intra-device reliability as an aim, which has been evaluated on a single participant. What is the generalizability of such results? You may consider this as “exploratory”

- Correlation analysis and interpretation of correlation outputs is misleading. Please remove it.

- “Our results suggest inaccurate sleep stage detection by GV4, but sleep onset and sleep end were accurately detected with few outliers.” The absence of a significant bias between PSG and device does not specifically inform about accuracy/inaccuracy.

- I would not consider low- vs high-frequency HRV as sympathetic vs vagal influence. I would quickly describe the main autonomic changes occurring across stages of sleep.

- It is unclear how the lights-off and lights-on were identified. Participant were asked to press an event-marker button when going to bed. How the authors determined the next morning wake time?

- What was the rate of concordance between the 2 scorers (MHL and TWK)?

- Please provide the data collection time windows (e.g., from Aug 2018 to …)

- Did the authors control for normality of data distribution in the analyses?

- The SD of WASO for GV4 is much lower than the SD of WASO for PSG. Please comment on that.

- Please provide BA plots for all the sleep parameters analyzed

- In the BA plots please use the PSG values on the x axis, and plot the bias or the proportional bias

- EBE should be performed on an individuals’ level. Sensitivity and specificity should be calculated for each participant (night) separately.

- Sensitivity and specificity should be provided for all sleep stages (separately)

- Can you please clarify this statement? “The high agreement in HR and movement detection between devices indicates differences in software rather than hardware”?

- The advantages of incorporating HR and HRV indices by these devices should be due to the sleep stage dependent shifting in autonomic control. Not sure what authors imply with this statement “HRV is influenced by several other conditions than sleep for instance stress level”.

- I would avoid any interpretation about GV4 performance due to the limited generalizability of the results (limitation of the sample used) and the lack of standard in evaluating what could be considered acceptable or not.

- Please check refs. 4, 6, 20 for accuracy

6. PLOS authors have the option to publish the peer review history of their article (what does this mean?). If published, this will include your full peer review and any attached files.

Reviewer #1: No

Reviewer #2: No

---

## [Author Response · Author response to Decision Letter 0]

18 Sep 2020

Responses to reviewer and editor comments are fully provided in the attached "Response to Reviewers" document (rebuttal letter).

---

## [Decision Letter · Decision Letter 1]

8 Oct 2020

PONE-D-20-21216R1

Assessing the performance of a commercial multisensory sleep tracker

PLOS ONE

Dear Dr. Mouritzen,

Thank you for submitting your manuscript to PLOS ONE. After careful consideration, we feel that it has merit but does not fully meet PLOS ONE’s publication criteria as it currently stands. Therefore, we invite you to submit a revised version of the manuscript that addresses the points raised during the review process.

As you can see from the attached comments, one of the reviewers has some remaining concerns that need to be addressed.

We look forward to receiving your revised manuscript.

Kind regards,

Raffaele Ferri, MD

Academic Editor

PLOS ONE

Reviewers' comments:

Reviewer's Responses to Questions

**Comments to the Author**

1. If the authors have adequately addressed your comments raised in a previous round of review and you feel that this manuscript is now acceptable for publication, you may indicate that here to bypass the “Comments to the Author” section, enter your conflict of interest statement in the “Confidential to Editor” section, and submit your "Accept" recommendation.

Reviewer #2: (No Response)

2. Is the manuscript technically sound, and do the data support the conclusions?

Reviewer #2: Yes

3. Has the statistical analysis been performed appropriately and rigorously? 

Reviewer #2: N/A

4. Have the authors made all data underlying the findings in their manuscript fully available?

Reviewer #2: No

5. Is the manuscript presented in an intelligible fashion and written in standard English?

Reviewer #2: Yes

6. Review Comments to the Author

Reviewer #2: The authors did a nice job in reviewing the paper and replying to the Reviewers’ concerns!

Few additional comments

- I would avoid statements like “Our results suggest inaccurate sleep stage detection by GV4”, etc. There is no standard on what would be considered accurate/inaccurate, and the picture is extremely complex (there are biases, proportional biases, clinical significance other than statistical significance to consider, etc.) . Thus, I believe that statements like that give the readers false interpretation of the device performance.

- In the intro, you may want to mention that PPG-derived HR and HRV has been “validated” against ECG-derived HR and HRV.

- Given the nature of the study you may want to specify which EEG derivation were used.

- I have trouble understanding the sleep onset and sleep end analysis. I usually consider SO the time in which the person falls asleep (the time of the first epoch of sleep after lights-off). Is that the case? What is the PSG event mark?

7. PLOS authors have the option to publish the peer review history of their article (what does this mean?). If published, this will include your full peer review and any attached files.

Reviewer #2: No

---

## [Author Response · Author response to Decision Letter 1]

16 Nov 2020

Our responses to specific reviewer and editor comments are provided in the attached file "Response_to_Reviewers_V1.2".

---

## [Editor Report · Decision Letter 2]

18 Nov 2020

Assessing the performance of a commercial multisensory sleep tracker

PONE-D-20-21216R2

Dear Dr. Mouritzen,

We’re pleased to inform you that your manuscript has been judged scientifically suitable for publication and will be formally accepted for publication once it meets all outstanding technical requirements.

Kind regards,

Raffaele Ferri, MD

Academic Editor

PLOS ONE
---

## [Editor Report · Acceptance letter]

26 Nov 2020

PONE-D-20-21216R2 

Assessing the performance of a commercial multisensory sleep tracker 

Dear Dr. Mouritzen:

I'm pleased to inform you that your manuscript has been deemed suitable for publication in PLOS ONE. Congratulations! Your manuscript is now with our production department. 

Kind regards, 

on behalf of

Dr. Raffaele Ferri 

Academic Editor

PLOS ONE